# Nutrients and Main Secondary Metabolites Characterizing Extracts and Essential Oil from Fruits of *Ammodaucus leucotrichus* Coss. & Dur. (Western Sahara)

**DOI:** 10.3390/molecules27155013

**Published:** 2022-08-06

**Authors:** Mohamed Lamin Abdi Bellau, Matteo Andrea Chiurato, Annalisa Maietti, Giancarlo Fantin, Paola Tedeschi, Nicola Marchetti, Massimo Tacchini, Gianni Sacchetti, Alessandra Guerrini

**Affiliations:** 1Pharmacist of the Sahrawi Refugee Camps, 37000 Tindouf, Algeria; 2Pharmaceutical Biology Lab., Research Unit 7 of Terra&Acqua Tech Technopole Lab., Department of Life Sciences and Biotechnology, University of Ferrara, Malborghetto di Boara, 44123 Ferrara, Italy; 3Department of Chemistry, Pharmaceutical and Agricultural Sciences, University of Ferrara, 44121 Ferrara, Italy

**Keywords:** *Ammodaucus leucotrichus*, proximate analysis, mineral element analysis, decoction, alcoholic extract, essential oil, secondary metabolites, ammolactone-A, R-perillaldehyde, chemical characterization

## Abstract

The ethnobotany of the Sahrawi people considers various species of plants and crude drugs as food, cooking spices and traditional health remedies. From among these, the fruits of *Ammodaucus leucotrichus* Coss. & Dur. (Apiaceae), known as Saharan cumin, were chosen for our research. The present paper reports a proximate composition and mineral element analysis of various samples of *A. leucotrichus* fruits, collected during the balsamic period (full fruiting) from plants grown in Bir Lehlu (Western Sahara) and purchased in a local market (Tindouf). These analyses pointed out interesting nutritional values of the crude drug. Decoction and alcoholic extract, analyzed by HPLC-DAD, evidenced ammolactone-A and R-perillaldehyde as the two main isolated constituents, particularly in the ethanolic extracts (ammolactone-A, market sample: 51.71 ± 0.39 mg/g dry extract; wild sample: 111.60 ± 1.80 mg/g dry extract; R-perillaldehyde, market sample: 145.95 ± 0.35 mg/g dry extract; wild sample: 221.40 ± 0.30 mg/g dry extract). The essential oils, obtained through hydrodistillation, were characterized by GC-MS and evidenced R-perillaldehyde (market sample: 53.21 ± 1.52%; wild sample: 74.01 ± 1.75%) and limonene (market sample: 35.15 ± 1.68%; wild sample: 19.90 ± 1.86%) as the most abundant compounds. The R configuration of perillaldehyde was ascertained and a complete description of the ^1^H and ^13^C NMR spectra of ammolactone-A was performed.

## 1. Introduction

The majority of the Sahrawi population has been living as refugees in the Hammada Desert in the southwestern part of Tindouf province (Algeria) for nearly 50 years. The refugees remain highly dependent on humanitarian aid for their survival, which has been reduced in recent years, causing a decrease in the distribution of fresh products, animal protein, etc., and impacting the population’s nutritional intake. The 2016 nutrition surveys revealed an improvement in some indicators, such as a decrease in global acute malnutrition (GAM) (2010), but a significant increase in others, such as the prevalence of anemia in pregnant (60%) and lactating women (72%), and in children under five years of age (28.4%). Reduced humanitarian funding is accompanied by the limited availability of food from local agricultural production. Although several international projects have encouraged agriculture aimed at diversifying food needs in refugee camps, the results are still marginal [1,2].

In addition to malnutrition and anemia, there are several diseases related to refugee status such as dental disease, and gastrointestinal or respiratory infections, with a high seasonal and environmental component, and a significant increase in non-communicable ailments, such as cardiovascular and thyroid ailments, hypertension, asthma and diabetes, which mainly affect the adult population which does not enjoy healthy ageing. In this context, it should be remembered that Saharawi traditional medicine uses local medicinal plants for the treatment of various diseases, such as gastrointestinal, airways respiratory tract disorders and diabetes [3].

Our research work fits into this complex scenario. Starting from the study of the local plants widely used in the Saharawi tradition for food and the treatment of diseases, and also through interviews with female healers, our research focused on *A. leucotrichus* Coss. & Dur. (Apiaceae), an endemic plant in North Africa, especially in the southern Algerian Sahara and Tassili regions [4], whose fruits are widely used in traditional medicine and in culinary recipes. For this reason, this crude drug, or its infusion and/or decoction as traditional preparations, could be a potential source of nutritional elements and useful metabolites for the treatment of disorders related to unbalanced nutrition. 

*A. leucotrichus*, known in Algeria as “Kammûn es-sofi” and “hairy cumin”, is an aromatic small annual species, glabrous and having a characteristic smell reminiscent of *Cuminum cyminum* L. The fruit is a diachene 8–10 mm long and is covered with very dense and fuzzy hairs. In the literature, the essential oil obtained from *A. leucotrichus* fruits has mainly been studied for its phytochemical composition and antioxidant, antimicrobial and anti-inflammatory activities [5]. The main component, perillaldehyde, appears to have the S absolute configuration [6]. Among the main secondary metabolites are documented ammolactone-A [7] and flavonoid derivatives of which luteolin-O-(malonylglucoside) is the most abundant [3]. The main macroelements are K (2283 mg/100 g dw) and Ca (1555 mg/100 g dw), while Fe (22 mg/100 g dw) shows the most relevant result among the microelements. The fruits also contain a good amount of lipids (11 g/100 g dw), characterized by the presence of linoleic acid (C18:2n6), oleic acid (C18:1n9), α-linolenic acid (C18:3n3) and palmitic acid (C16:0) as the predominant fatty acids [3].

This research work has defined the absolute configuration of perillaldehyde and performed the complete description of the ^1^H and ^13^C NMR spectra of ammolactone-A, filling in the limited literature data. For the first time, perillaldehyde and ammolactone-A were detected and quantified in decoction and alcoholic extracts of *A. leucotrichus* fruits and prepared using wild-collected samples and market-purchased ones. Finally, a comparison with literature data was performed for nutritional parameters of the fruits and essential oils obtained from them by hydrodistillation.

## 2. Results and Discussion

### 2.1. Proximate Analysis and Mineral Composition

The proximate analysis, carried out on the fruits of *A. leucotrichus* (Table 1), showed quite similar residual moisture content in both samples, around 10–12%, consistent with the typical moisture content of dried aromatic plants [8]. The protein contents were also comparable (10.77 ± 0.07 g/100 g and 9.14 ± 0.62 g/100 g) for both samples but lower than in the literature data, while the lipids were almost three times more in the sample harvested in Bir Lehlu, which was similar to previously published data, than in those purchased in the Tindouf market (11.30± 0.14 g/100 g vs 4.05 ± 0.01 g/100 g); in both cases, these values are clearly lower than those of cumin, a spice used for culinary preparations and also belonging to the Apiaceae family [9], of which *A. leucotrichus* can be considered as a succedaneum.

As far as fiber analysis is concerned, *A. leucotrichus* not only showed a high content of this nutrient for both samples (74.51 ± 2.04 g/100 g of total fiber for the market sample, 72.00 ± 3.55 g/100 g for wild-collected sample) but, above all, the percentage of insoluble fiber was found to be more concentrated than the soluble one, especially for the market sample (66.68 ± 1.66 g/100 g). This result is very interesting because high-fiber diets could be associated with the prevention and treatment of certain diseases such as diabetes, also common among the Saharawi population; consequently, this plant could be considered a potential nutraceutical ingredient for the prevention and treatment of this disease [11].

Concerning the analysis of minerals (Table 2), both samples showed a high concentration of K and Ca content quite similar to cumin, but lower than data previously published for *A. leucotrichus* fruits. The health benefits of potassium could be relevant for blood pressure, bone density and risk of kidney stones. Calcium is an important macroelement involved in the regulation of muscle contraction, blood coagulation, the transmission of nerve impulses, the regulation of cell permeability, the activity of numerous enzymes (promoting the release of insulin by pancreatic cells) and the growth and fortification of teeth and bones [12,13]. The samples highlighted a high iron content (132.0 ± 4.4 mg/100 g and 86 ± 11 mg/100 g, respectively, for market and wild-collected samples), and good zinc (1.64 ± 0.02 mg/100 g and 2.98 ± 0.29 mg/100 g, respectively, for market and wild-collected samples) and manganese (3.84 ± 0.04 mg/100 g and 2.93 ± 0.01 mg/100 g, respectively, for market and wild-collected samples) values. With particular reference to iron, the data of our samples were more interesting than those previously published for *A. leucotrichus* and *C. cyminum*.

These results have importance especially when compared with the nutritional values of cumin; in fact, the high content of such minerals could help in reducing problems related to their deficiency, such as growth delay and anemia, which have a significant incidence in the Saharawi population.

Regarding the fatty acids profile (Table 3), both samples showed a high percentage of oleic acid (86.07% and 86.7%, respectively, for market and wild-collected samples) and a lower percentage of linoleic acid (8.33% and 9.27%), with the oleic-linoleic ratio > 7. This profile recalls that of olive oil, a lipidic food with a composition most similar to endogenous nutritional needs. Several researchers have suggested that its consumption is associated with a reduced risk for several chronic illnesses, such as diabetes, hypertension, obesity and cardiovascular diseases. Furthermore, the fatty acids profile of *A. leucotrichus* presented a percentage of palmitic acid lower than olive oil (7–17%). Foods with a high content of saturated fatty acids, such as palmitic acid, are considered to be among the factors preventing cardiovascular diseases [14,15]. The fatty acids profile of our samples was different from previously published data, especially with reference to the percentage of palmitic acid, linoleic and α-linolenic acid; on the other hand, it seemed to be very similar in composition to cumin.

### 2.2. Chemical Characterization of A. leucotrichus Essential Oil, Extracts and Their Main Components

The composition of the essential oils of *A. leucotrichus* is shown in Table 4; the main compounds identified belong to the class of oxygenated monoterpenes, including R-perillaldehyde as a major component (53.31% for market sample, 74.01% for wild sample), followed by hydrocarbon monoterpenes, with limonene as the main compound (35.15% for market sample, 19.90% for wild sample).

A comparison of the percentages of the main compounds (limonene and R-perillaldehyde) revealed in *A. leucotrichus* essential oil in previous papers is summarized in Table 5 and shows that our results are very similar to those of the literature in terms of quality, but from a quantitative point of view there are remarkable differences, probably due to the different harvesting period, latitude, climate and soil type. In fact, the content of perillaldehyde fluctuates between a maximum of 84.4% and a minimum of 53.2%, while that of limonene is between 35.15% and 1.7%.

Interestingly, the composition of the essential oil of *A. leucotrichus* is very similar to that of the essential oil of another Iranian plant, namely *Dracocephalum surmandinum*, where the two main components of the volatile fraction are characterized by perillaldehyde (54.3%) and limonene (30.1%) [22], and of the essential oil of leaf *Perilla frutescens* (L.) Britt., the best-known source of S-(−)-perillaldehyde [23].

The yield of essential oils (Table 4) was respectively 2.00 ± 0.02% for the market sample and 3.80 ± 0.06% for the wild-collected sample, representing the best value with respect to the literature data.

The isolation of the molecules, perillaldehyde and ammolactone-A, from the ethanol extract, allowed them to be chemically characterized. The GC-MS of perillaldehyde showed a purity of 96% and an experimental mass fragmentation comparable to literature data [16] (*m*/*z*): 151 (15), 150 (15), 135 (50), 122 (45), 107 (70), 93 (60), 91 (75), 79 (100), 68 (55), 67 (100), 77 (45), 53 (30). As reported by Chebrouk et al. (2019) [6], this molecule in *A. leucotrichus* had the absolute configuration of S-(−)-perillaldehyde. The injection of isolated perillaldehyde and commercial S-(−)-perillaldehyde (Sigma Aldrich, Burlington, MA, USA) in GC-FID, equipped with chiral column, highlighted that the isolated perillaldehyde is not superimposable with the commercial one and corresponded to enantiomer R (Figure 1), as we recently reported [24]. The findings are in contrast to previous literature data in which the absolute configuration S has been supposed without experimental evidence. In addition, the optical rotation was: ${\left[\alpha \right]}_{D}^{20}$ = +115 (c 10, ethanol), opposite to the commercial S-(−)-perillaldehyde. 

Through GC-MS analysis we established that ammolactone-A had a purity of 97% and an experimental mass fragmentation comparable to literature data [7] (*m*/*z*): 332 (1), 293 (3), 248 (20), 233 (5), 230 (25), 205 (20), 190 (30), 175 (35), 169 (10), 167 (20), 159 (50), 157 (50), 145 (45), 133 (35), 132 (75), 120 (45), 107 (35), 105 (35), 85 (25), 81 (65), 79 (65), 57 (100). The same authors described only partially the NMR spectra of this molecule. For this reason, we performed the complete description of its ^1^H and ^13^C NMR spectra. ^1^H NMR (400MHz, CDCl_3_): 5.52 (1H, m, H-3), 5.42 (1H, ddd, J = 11.2 Hz; J = 9.4 Hz; J = 0. 8 Hz, H-8), 4.57 (1H, dd, J = 11,6 Hz, J = 9.1 Hz, H-6), 3.05 (1H, ddd, J = 11.2 Hz; J = 9.3 Hz; J = 9.2 Hz, H-7), 2.70 (1H, dq, J = 9.3 Hz; J = 7.8 Hz, H-11), 2.62 (1H, brs, OH-10), 2.60 (1H, dd, J = 11.4 Hz; J = 5.6 Hz, H-5), 2.41 (1H, m, H-1), 2.37 (1H, sex, J = 7.0 Hz, H-17), 2.18 (1H, m, H-2), 2.09 (1H, m, H-2′), 2.07 (1H, dd, J = 14.7 Hz, J = 9.6 Hz, H-9), 1.87 (3H, dt, J = 2.8 Hz; J = 1.5 Hz, CH_3_ C-4), 1.75 (1 H, dd, J =14.7 Hz; J = 1.8 Hz; J = 0.8 Hz, H-9′), 1.68 (1H, dquint, J = 13.6 Hz, J = 7.4 Hz, H-18), 1.48 (1H, dquint, J = 13.6 Hz; J = 7.4 Hz, H-18′), 1.31 (3H, d, J = 7.8 Hz, CH_3_ C-13), 1.22 (3H, s, CH_3_ C10), 1.14 (3H, d, J = 7.0 Hz, CH_3_ C-17), 0.92 (3H, t, J = 7.4 Hz, CH_3_ C-18). 

^13^C NMR (400 MHz, CDCl_3_): 179.4 (C12), 177.4 (C16), 147.2 (C4), 125.4 (C3), 80.8 (C6), 71.6 (C10), 67.3 (C8), 55.7 (C17), 50.2 (C5), 45.6 (C7), 43.3 (C9), 41.3 (C1), 36.3 (C11), 32.4 (C2), 31.3 (C14), 26.8 (C18), 18.9 (C15), 16.6 (C20), 13.6 (C13), 11.8 (C19).

All the proton resonances were associated with those of the directly attached carbon atoms through the DEPT and 2D NMR HMQC experiments. The proton multiplets were arranged in sequence through the COSY experiment, yielding the three spin systems evidenced in bold in Figure 2. The key HMBC correlations, marked with the arrows in Figure 2, detect the heteronuclear correlations of the quaternary carbons C(4), C(10), C(12) and C(16) of ammolactone-A.

As shown in Table 6, the ethanol extract from the wild-collected sample evidenced a higher amount of both ammolactone-A and R-perillaldehyde, the two main compounds detected. The decoction had a lower content of these two metabolites. It should be emphasized that there are no references in the literature concerning the characterization of these preparations.

## 3. Materials and Methods

### 3.1. Plant Material

The fruits of *Ammodaucus leucotrichus* Coss. & Dur. were collected from a wild population of plants at Bir Lehlu (coordinates: 26°20′58″ N 09°34′32″ W, Western Sahara) and purchased in Tindouf local market. The samples authentication was performed by Dr. Mohamed Lamin Abdi Bellau and Prof. Alessandra Guerrini through the IUCN Centre For Mediterranean Cooperation (2005) [25]. Wild plant material was dried at room temperature for 15 days. The voucher specimen (code no. AMM.022.001) is stored in the Herbarium of the University of Ferrara (Italy). The present research is compliant with the Nagoya protocol.

### 3.2. Proximate Analysis

To determine the moisture, the samples were finely ground with a knife mill (Grindomix GM200, Retsch, Dusseldorf, Germany), then dried in an oven at 110 °C until a constant weight. Moisture was expressed as g/100 g.

Proteins (total nitrogen compounds) were determined on 1 g of dry matter, according to the Kjeldahl method [26]. The protein content was determined by means of a conversion factor equal to 6.25. 

Total mineral contents (total ashes) were determined on 1 g of dry matter, in a muffle furnace at 570 °C overnight. Total ashes were expressed as g/100 g.

To determine the total lipidic content, about 3 g of dried crude drug was put in a thimble and introduced into the Soxhlet extraction unit (VelpScientifica, Usmate, Milan, Italy) with 50 mL of diethyl ether (CarloErba, Rodano, Milan, Italy). The lipids were dissolved in 3 mL of hexane and transesterified with 1.5 mL of 5% sodium hydroxide in methanol to obtain free fatty acid methyl esters. Sample volume of 1 μL was injected into the gas chromatography-mass spectrometry (Varian Saturn 2100 MS/MS ion trap mass spectrometer coupled to a Varian 3900 gas-chromatograph). The separation was achieved with a capillary column Zebron ZB-WAX (Phenomenex) (60 m, 0.25 mm i.d., 25 µm film thickness) supplied with helium carrier gas at 1 mL/min constant flow. The injector temperature was 260 °C and the oven temperature program was the following: start 100 °C for 2 min, ramp to 200 °C at 10 °C/min and hold for 58 min. 

Dietary fiber (total, soluble and insoluble) was determined on 1 g of duplicate dried samples as per the instruction protocol developed for the Megazyme Total Dietary Fiber Assay Kit (Megazyme International, Co., Wicklow, Ireland) and based on the combined enzymatic and gravimetric method of Lee et al. (1992) [27] and Prosky et al. (1988) [28].

### 3.3. Mineral Composition Determination

Na, K, Mg, Zn, Ca, Fe, Cu and Mn were determined according to previously described methods [29]. One gram of dry matter was mineralized in CEM’s digestion vessels (PTFE mod. SV140, FKV) with HNO_3_-H_2_O_2_ in an oven (DK6 heating digester, VELP) with a temperature program and coupled with a module for steam extraction (EM 5, FKV).

A Perkin–Elmer (Perkin-Elmer, Inc.,Waltham, MA, USA) (mod. 1100B) atomic absorption spectrometer (AAS) was used for the analysis. The spectrometer was equipped with a deuterium background corrector and single-element Intensitron (Perkin–Elmer, Inc.,Waltham, MA, USA). Standard solutions of each element were prepared by diluting reference standard solutions for AAS (BDH certified atomic absorption reference solutions). The samples were checked against reference standards and measured for their absorbance after instrument calibration. An average of five readings of absorbance was taken for all samples.

### 3.4. Hydrodistillation of A. leucotrichus Fruits

The pale blue essential oils, with a characteristic odor, were obtained performing a standardized hydrodistillation process, optimized for *A. leucotrichus* crude drug, that used 20 g of dried fruits in 500 mL of water in a Clevenger-type apparatus for 3 h. Three separate distillations were performed for each sample and successively the obtained essential oils were pooled, dried over anhydrous sodium sulfate (Na_2_SO_4_) and stored at 4 °C in amber glass vials until analysis [30].

### 3.5. Preparation of Ethanolic Extract and Decoction

An aliquot of 4.0 g of shredded fruits of *A. leucotrichus* was extracted with 200 mL of ethanol by ultrasound assisted maceration (Ultrasonik 104X, Ney Dental International, MEDWOW, Nicosia, Cyprus), total volume 10.4 L, internal dimensions: 146 × 292 × 241 mm, frequency analysis: 48 kHz) for 1 h at 25 °C. Three distinct extractions were performed for each sample. The combined extracts were then filtered and concentrated with a rotary evaporator (RV 10 digital, IKA^®^-Werke GmbH & Co. KG, Staufen im Breisgau, Germany). 

To 2.5 g of shredded fruits of *A. leucotrichus* was added 50 mL of distilled water. The mixture was placed on a heating plate under magnetic stirring and once it reached 100 °C it was left in such conditions for 15 min [31]. Three distinct extractions were performed for each sample. The combined decoctions were then filtered and lyophilized.

### 3.6. Isolation of Chemical Constituents and Quantitative Chemical Characterization of Extracts and Essential Oil

The isolation of unknown compounds was performed in a silica gel chromatographic column (silica gel 60 220–440 mesh, particle size: 0.035–0.070 mm, Sigma-Aldrich, Milan, Italy). An aliquot of 500 mg of ethanolic extract was suspended in 2 mL of mobile phase: hexane: ethyl acetate (8:2). Precoated silica gel plates (silica gel 60 F_254_; thickness 0.25 mm; Merck, Milan, Italy) with the same above mobile phase were used to control the fraction separations; after development, the plate was sprayed with phosphomolybdic acid solution (20% phosphomolybdic acid in EtOH) [32] and heated to 120 °C. The isolated molecules showed an intense blue color on a yellow background. The solvents of collected fractions were evaporated to dryness with a rotary evaporator (RV 10 digital, IKA^®^-Werke GmbH & Co. KG, Staufen im Breisgau, Germany). The isolation by silica gel column gave 40 mg of perillaldehyde and 70 mg of ammolactone-A. White crystals of ammolactone-A were recrystallized from hot cyclohexane. 

The GC-MS was used to analyze the essential oil, the identity and the purity of the separated molecule. The GC-MS analysis was performed with a Varian 3800 chromatograph (Varian, Palo Alto, CA, USA) equipped with a Varian Factor Four VF-5 ms column (5%-phenyl-95%-dimethylpolysiloxane, internal diameter: 0.25 mm, length: 30 m) interconnected with a Varian mass spectrometer SATURN MS-4000 (Varian,, Palo Alto, CA, USA), with electronic impact ionization, ion trap analyzer and software provided with the NIST database for the identification of components. The experimental conditions used were the following: helium carrier gas (1 mL/min), a split ratio of 1:50, ionization energy (EI) 70 eV, emission current of 10 μA, scan rate of 1 scan/sec, mass range 40–400 Da. For the analysis, the oven initial temperature of 70 °C was increased to 230 °C with a rate of 4 °C/min and maintained at 230 °C for 10 min; finally, it was brought from 230 to 280 °C with an increase of 5 °C/min. The total time of acquisition of the chromatogram was 70 min. The arithmetic index of perillaldehyde was determined by adding a C_8_–C_32_ n-alkanes (Sigma-Aldrich) mixture to the essential oil before injecting into the GC–MS equipment, following the same conditions reported above [33].

The pure perillaldehyde was compared with the commercial S-(−)-perillaldehyde (Sigma Aldrich) through GC analyses, performed with a Thermo Focus-gas chromatograph equipped with a flame ionization detector and a chiral Megadex 5 column (25 m × 0.25 mm), with the following temperature program: 80 to 200 °C, rate 2 °C min^–1^. Optical rotation was measured at 20 ± 2 °C in ethanol, 10% concentration; ${\left[\alpha \right]}_{D}^{20}$_D_ value is given in 10^−1^deg cm^2^ g^−1^.

The ^1^H, ^13^C, DEPT NMR and the 2D NMR experiments (COSY, HMQC, HMBC) spectra for the characterization of ammolactone-A were recorded with a Varian Mercury Plus 400, operating at 400 MHz (^1^H) and 100 MHz (^13^C), respectively. The chemical shifts were referenced to the residual solvent signal (CDCl_3_: δ_H_ 7.26, δ_C_ 77.16). The 2D NMR experiments (COSY, HMQC, HMBC) were processed using the MestReNova (Santiago de Compostela, Spain) Version 6.0.2-5475 software.

HPLC analyses of ethanolic extract and decoction were performed using JASCO modular HPLC system (Tokyo, Japan, model PU 2089) coupled to a diode array apparatus (MD 2010 Plus) linked to an injection valve with a 20 μL sampler. A Zorbax-Eclipse plus-C18 (250 × 4.6 mm, 5 µm) was used at a flow rate of 1.0 mL/min. The mobile phase consisted of solvent solution A (water/phosphoric acid pH = 2.9) and B (acetonitrile). The adopted gradient system consisted of the following steps: isocratic condition of 75:25 *v*/*v* (A/B) until 10 min; gradual changing to 20:80 *v*/*v* up to 30 min; progressive raise to 0:100 *v*/*v* up to 35 min and isocratic mode up to 45 min; back to starting point (75:25 *v*/*v*) in 10 min. The injection volume was 40 μL. The chromatograms were observed at 190 nm. For perillaldehyde the calibration range was 500–10 µg/mL, the correlation coefficient (r^2^) 0.993, the limit of quantification (LOQ) 7 μg/mL, the limit of detection (LOD) 2 μg/mL; for ammolactone-A the calibration range was 500–2.5 µg/mL, the correlation coefficient (r^2^) 0.999, the limit of quantification (LOQ) 8 μg/mL, the limit of detection (LOD) 3 μg/mL.

## 4. Conclusions

The crude drug of *A. leucotrichus* is mainly known in the literature for its essential oil and its antimicrobial and anti-inflammatory activities, mainly related to the presence of perillaldehyde: in this work, we established the absolute configuration of the molecule (R), which differs from that found in *P. frutescens* (S). We also provided the NMR characterization of ammolactone-A, which is present, together with perillaldehyde, in the alcohol extract and decoction we prepared from two samples, one collected in the wild and the other purchased in a market. These two secondary metabolites were quantified in these two preparations. The essential oil obtained by hydrodistillation had R-perillaldehyde and limonene as the main components.

The samples showed a high fiber content, which could be useful for the prevention and treatment of certain diseases such as diabetes, which is common among the Saharawi population. The high content of Ca, K and Fe could help in reducing problems related to deficiency in these minerals, such as growth delay and anemia, which also have a strong incidence in the Saharawi population.

## Figures and Tables

**Figure 1 molecules-27-05013-f001:**
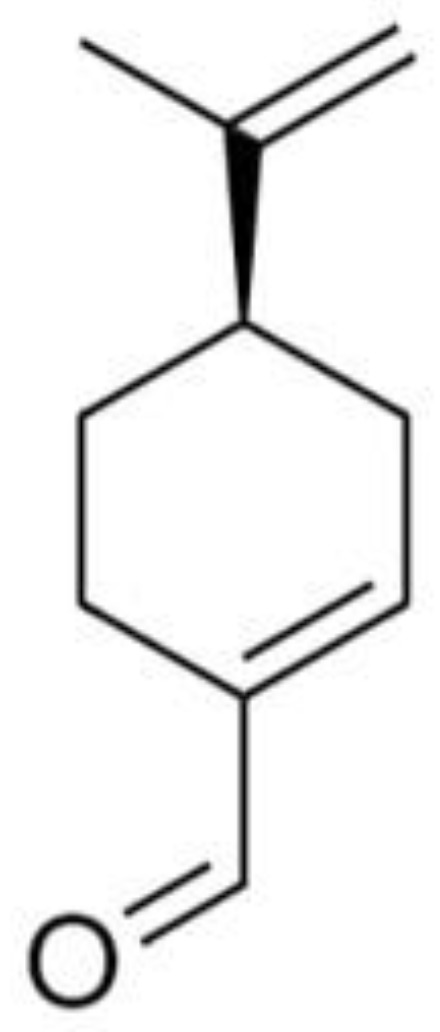
R-perillaldehyde.

**Figure 2 molecules-27-05013-f002:**
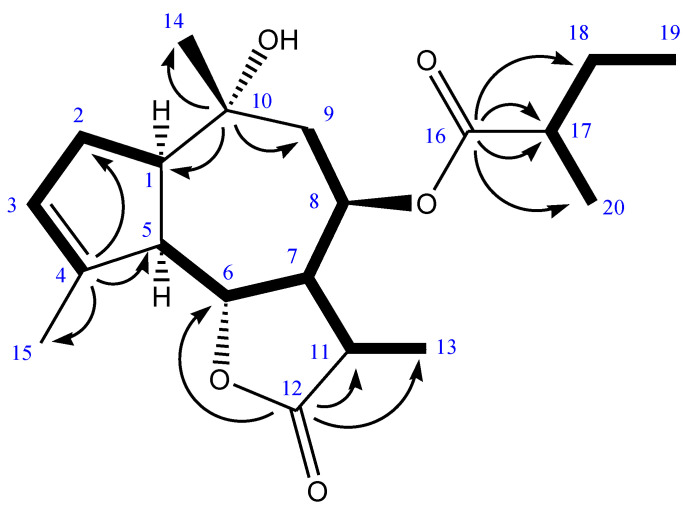
Proton connectivity network by COSY (bold lines) and key HMBC cross-peaks of ammolactone-A. Further HMBC cross-peaks are omitted for clarity.

**Table 1 molecules-27-05013-t001:** Proximate analysis of *A. leucotrichus* samples compared to literature data for *A. leucotrichus* and *C. cyminum*.

Proximate Analysis	*A. leucotrichus* Market	*A. leucotrichus* Wild	*A. leucotrichus* ^1^	*Cuminum* *cyminum* ^2^
Humidity (g/100 g)	12.65 ± 0.07	10.59 ± 0.33	/	6.44
Proteins (g/100 g)	10.77 ± 0.07	9.14 ± 0.62	13.1 ± 0.9	19.00
Lipids (g/100 g)	4.05 ± 0.01	11.30 ± 0.14	11.1 ± 0.3	29.17
Total ash (g/100 g)	14.43 ± 0.02	10.00 ± 0.01	10.8 ± 0.2	7.36
Total fiber (g/100 g)	74.51 ± 2.04	72.00 ± 3.55	/	51.3
Insoluble fiber (g/100 g)	66.68 ± 1.66	63.85 ± 4.01	/	46.4
Soluble fiber (g/100 g)	7.83 ± 0.38	8.15 ± 0.45	/	4.89

^1^ Literature data [3]. ^2^ Literature data [9,10].

**Table 2 molecules-27-05013-t002:** Mineral determination of *A. leucotrichus* samples compared to literature data for *A. leucotrichus* and *C. cyminum*.

Minerals	*A. leucotrichus* Market	*A. leucotrichus* Wild	*A. leucotrichus* ^1^	*Cuminum cyminum* ^2^
Macroelements				
Na (mg/100 g)	110 ± 19	156 ± 4	160 ± 6	168
Mg (mg/100 g)	254 ± 36	234 ± 47	236.6 ± 0.4	337
K (mg/100 g)	1949 ± 99	1636 ± 88	2283 ± 0.4	1790
Ca (mg/100 g)	737 ± 10	691 ± 48	1555 ± 2	917
Microelements				
Fe (mg/100 g)	132.0 ± 4.4	86 ± 1.1	22 ± 2	14.0
Zn (mg/100 g)	1.64 ± 0.02	2.98 ± 0.29	1.72 ± 0.04	3.0
Cu (mg/100 g)	0.68 ± 0.03	0.64 ± 0.01	0.39 ± 0.03	0.87
Mn (mg/100 g)	3.84 ± 0.04	2.93 ± 0.01	7.6 ± 0.4	3.46

^1^ Literature data [3]. ^2^ Literature data [9,10].

**Table 3 molecules-27-05013-t003:** Fatty acids (relative percentage) of *A. leucotrichus* samples compared to literature data for *A. leucotrichus* and *C. cyminum*.

Fatty Acid	*A. leucotrichus* Market	*A. Leucotrichus* Wild	*A. leucotrichus* ^1^	*Cuminum cyminum* ^2^
C14:0 (myristic)	0.06	<0.05	0.87	0.10
C16:0 (palmitic)	3.83	3.33	21.2	6.13
C16:1 (palmitoleic)	0.40	0.33	1.06	1.13
C18:0 (stearic)	1.10	<0.05	0.10	1.83
C18:1n9c (oleic)	86.07	86.7	53.8	73.17
C18:2n6c (linoleic)	8.33	9.27	1.99	16.68
C18:3n3 (α-linolenic)	0.22	0.38	10.6	0.97

^1^ Literature data [3]. ^2^ Literature data [10].

**Table 4 molecules-27-05013-t004:** Essential oil composition of *A. leucotrichus* samples.

No.	Component ^1^	*A. leucotrichus* Market Area% ^2^	*A. leucotrichus* Wild Area% ^2^	AI Exp ^3^	AI Lit ^4^
1	α-pinene	0.77 ± 0.02	1.28 ± 0.02	928	932
2	camphene	3.33 ± 0.12	0.09 ± 0.01	943	946
3	β-pinene	0.33 ± 0.04	0.90 ± 0.04	972	974
4	myrcene	-	0.21 ± 0.02	986	988
5	p-mentha-1(7),8-diene	0.16 ± 0.01	-	1001	1003
6	3-carene	2.1 ± 0.03	1.15 ± 0.01	1005	1008
7	**limonene**	**35.15 ± 1.68**	**19.90 ± 1.86**	1024	1024
8	**R-perillaldehyde**	**53.21 ± 1.52**	**74.01 ± 1.75**	1272	1269
9	perillyl alcohol	2.41 ± 0.07	1.01 ± 0.11	1296	1294
10	methylperillate	0.71 ± 0.02	0.91 ± 0.02	1395	1392
11	*cis*-β-caryophyllene	0.22 ± 0.02	-	1400	1408
12	germacrene D	0.35 ± 0.01	-	1483	1485
13	δ-amorphene	-	0.10 ± 0.02	1514	1511
	Total identified	98.74	99.56		

^1^ Components are listed in order of elution and their nomenclature is in accordance with the NIST (National Institute of Standards and Technology) library. ^2^ Relative peak areas, calculated by GC-FID. ^3^ AI exp: linear retention indices calculated on Varian VF-5 ms column. ^4^ AI lit: linear retention indices [16]. The main compounds are in bold.

**Table 5 molecules-27-05013-t005:** Main components of *A. leucotrichus* essential oils in literature and their extraction yields.

Compound	*A. leucotrichus* Market (Area%)	*A. leucotrichus* Wild (Area%)	(Area%) ^A^	(Area%) ^B^	(Area%) ^C^	(Area%) ^D^	(Area%) ^E^
limonene	35.15	19.90	28.8	1.7	26.8	8.2	6.9–29.2
perillaldehyde	53.21	74.01	56.4	84.4	63.6	87.9	60.1–37.5
Yield	2.00%	3.80%	0.7%	2.58%	2.76%	1.6%	2.0–2.1%

A = [17]; B = [18]; C = [19]; D = [20]; E = [21].

**Table 6 molecules-27-05013-t006:** HPLC-DAD quantitative analysis of fruits extracts of *A. leucotrichus*.

Component	*A. leucotrichus* Market Decoction ^1^	*A. leucotrichus* Market Ethanolic Extract ^1^	*A. leucotrichus* Wild Decoction ^1^	*A. leucotrichus* Wild Ethanolic Extract ^1^
ammolactone-A	5.68 ± 0.12	51.71 ± 0.39	3.32 ± 0.22	111.6 ± 1.8
R-perillaldehyde	24.32 ± 0.93	145.95 ± 0.35	35.88 ± 0.60	221.4 ± 0.3

^1^ mg/g dry extract ± (sd).

## Data Availability

Data is contained within the article.

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
