# Peer review of "Nutrients and Main Secondary Metabolites Characterizing Extracts and Essential Oil from Fruits of Ammodaucus leucotrichus Coss. & Dur. (Western Sahara)"

_molecules, 2022, doi:10.3390/molecules27155013_

Round 1

Reviewer 1 Report

Nutrients and main secondary metabolites characterizing the extracts and essential oil from fruits of Ammodaucus leucotrichus Coss. & Dur. (Western Sahara) were investigated. In this work, the absolute configuration of the molecule perillaldehyde (R) was determined which differs from that found in P. frutescens (S). The NMR characterisation of amolactone-A, which is present, together with perillaldehyde, in the alcohol extract and decoction was performed. The essential oil obtained by hydrodistillation contained R-perillaldehyde and limonene as the main components.  The samples showed a high-fibre content. The high content of Ca, K, and Fe was determined.

The manuscript contains interesting data regarding absolute configuration determination and the structure elucidation using NMR. However, it needs major revision before it could be considered further.

The English grammar should be significantly improved through overall text.

Lot of general data is presented regarding different diseases and refugees, but this is not the focus of present research that is chemical study related to phytochemical profiling and determination of the compounds structures. With this respect the abstract, introduction (lines 36-55), and discussion part should be changed to contain less info about general diseases and biological activity that were not investigated in present research. Also connection among the compounds with known bioactivity with different diseases should be tested if discussed in detail, if not tested only probable bioactivity should be mentioned.

Particular remarks

 The abstract is too general and it needs to be corrected significantly. Lines 17-20 should be omitted since they present very general information and this is not the focus of present research. On the other hand, the results of decoction, alcoholic extract and essential oil obtained by hydrodistillation that were instead analyzed by GC-MS and HPLC-DAD should be reported with more details in the abstract. In particular, determination of the present enantiomer and NMR results should be presented.

Cuminum cyminum L. , Perilla frutescens should be written Italic throughout overall text.

 At the end of introduction part, lines 78-84, there is need to point out more clearly what is the novelty of present research in comparison to already published data on A. leucotrichus. What was done for the first time in present research?

 Lines 88-89 “…proving that the dehydration process of the plant doesn’t affect the fruits.” . It is not clear, what does it mean that the dehydration process of the plant doesn’t affect the fruits?

 How stereochemistry of D-limonene was established? If chiral column was not used than it it impossible to determine its stereochemistry and absolute configuration.

 Z-caryophyllene should be more precisely written with respect to the nomenclature.

 Determined stereochemistry of perillaldehyde should be marked in all the tables with the results.

 Paragraph 3.4 “20 g of dried fruits in 500 ml of water” – is it correct; it seems to high amount of water for hydrodistillation for 20 g of the sample.

Author Response

Dear Reviewer,

                                  thanking very much for your valuable suggestions, following are reported our replies point by point. The manuscript has also changed in accordance with what is reported below using the word changes tracking system.

General comments:

  • The English grammar should be significantly improved through overall text.

The English grammar was improved through overall text.

  • “…..abstract, introduction (lines 36-55), and discussion part should be changed to contain less info about general diseases and biological activity that were not investigated in present research. Also connection among the compounds with known bioactivity with different diseases should be tested if discussed in detail, if not tested only probable bioactivity should be mentioned.”

Thanking the referee for this criticism, we however believe that all the information on specific diseases of Sahrawi people and the biological activities reported by related literature are important because they are the premise and motivation of the research, then actually developed about the Ammodaucus leucotrichus medicinal crude drug. This aspect is further stressed in the introduction paragraph at the lines 66-73.

  • Regarding the suggestion “Also connection among the compounds with known bioactivity with different diseases should be tested if discussed in detail, if not tested only probable bioactivity should be mentioned.”

The text has been changed pointing out the “probable bioactivity” where appropriate. However, the statements about the correlation between bioactivities and diseases are reported in the results and discussion section always with reference to related literature.

Particular remarks

  • The abstract is too general and it needs to be corrected significantly. Lines 17-20 should be omitted since they present very general information, and this is not the focus of present research. On the other hand, the results of decoction, alcoholic extract and essential oil obtained by hydrodistillation that were instead analyzed by GC-MS and HPLC-DAD should be reported with more details in the abstract. In particular, determination of the present enantiomer and NMR results should be presented.

The abstract has been deeply changed and rewritten, omitting general information and focusing on the phytochemical profiling of the samples extracts, on the determination of the occurring enantiomer and NMR results.

  • Cuminum cyminum L., Perilla frutescens should be written Italic throughout overall text.

The scientific name of the plant species Cuminum cyminum and Perilla frutescens have been written in italic.

  • At the end of introduction part, lines 78-84, there is need to point out more clearly what is the novelty of present research in comparison to already published data on A. leucotrichus. What was done for the first time in present research?

The final part of the Introduction has been rewritten pointing out the novelty of our research.

  • Lines 88-89 “…proving that the dehydration process of the plant doesn’t affect the fruits.” . It is not clear, what does it mean that the dehydration process of the plant doesn’t affect the fruits?

Agreeing with the referee, the sentence has been deleted because it is a typographical error.

  • How stereochemistry of D-limonene was established? If chiral column was not used than it impossible to determine its stereochemistry and absolute configuration.

Agreeing with the referee, the D- configuration has been deleted throughout all the text for limonene. The D- was a typographical error.

  • Z-caryophyllene should be more precisely written with respect to the nomenclature.

The name of the compound has been corrected as “cis-ß-caryophyllene” 

  • Determined stereochemistry of perillaldehyde should be marked in all the tables with the results.

The determined stereochemistry has been specified where appropriate.

  • Paragraph 3.4 “20 g of dried fruits in 500 ml of water” – is it correct; it seems to high amount of water for hydrodistillation for 20 g of the sample.

The water amount is correct, and it is the result of the optimization of the hydro-distillation. This aspect has been specified in the text. However, as reported for some European Pharmacopoeia crude drugs, the same water amount is required for similar amount of plant material. 

Reviewer 2 Report

The present article reflects a rather simple but interesting work on the important Saharan nutritional plant Ammodaucus leucotrichus. Measurements of fibre, minerals and quality of lipids reflect this importance in terms of battling malnutrition, while a simple phytochemical work sheds light on the stereochemistry of isolated perillaldehyde, and the NMR of ammolactone-A. 

Major revisions: 

-Please include in the experimental part the voucher number of the plant specimen, and in which herbarium is stored. Please also refer to the compliance of the research to the Nagoya protocol

-Since the stereochemistry of perillaldehyde is revised, It is of essence to include in the manuscript a figure with the chromatogram on a chiral column comparing the two enantiomers of the isolated and the commercially available compound. In the figure, the structure of perillaldehyde must be included. 

Minor revisions

-Throughout the text, all plant species in italics

-Typographical errors such as: subscripts in abbreviations (CDCl3, CH3), etc

-Correct spelling of ammolactone-A throughout the text

Author Response

Dear Editor,

                                  thanking  for your valuable suggestions, following are reported our replies point by point. The manuscript has also changed in accordance with what is reported below using the word changes tracking system.

Major revisions:

  • Please include in the experimental part the voucher number of the plant specimen, and in which herbarium is stored. Please also refer to the compliance of the research to the Nagoya protocol

The voucher number of the plant specimen together with the herbarium in which the sample is stored has been added (section 3.1). A reference to the compliance of the research to the Nagoya protocol has been also added in the same section.

  • Since the stereochemistry of perillaldehyde is revised, It is of essence to include in the manuscript a figure with the chromatogram on a chiral column comparing the two enantiomers of the isolated and the commercially available compound. In the figure, the structure of perillaldehyde must be included.

Unfortunately, the figure referred to by the referee was already included in another manuscript submitted and its inclusion would infringe the copyright of the journal to which it was submitted. We have therefore cited in the text the following bibliographic reference to the manuscript that contains the figure to which the referee referred: Catanzaro, E.; Turrini, E.; Kerreb, T.; Sioen,  S.; Baeyens, A.; Guerrini, A.; Abdi Bellau, M.L.; Sacchetti, G.; Paganetto, G.;  V. Krysko, D.; Fimognari, C.  Perillaldehyde is a new ferroptosis inducer with a relevant clinical potential for acute myeloid leukemia therapy. Biomed.Pharmacother., submitted for publication.

As regard the structure of R-perillaldehyde, a proper figure has been added.

Minor revisions

  • Throughout the text, all plant species in italics.

The manuscript has been checked for the scientific name of the species in italics.

  • Typographical errors such as: subscripts in abbreviations (CDCl3, CH3), etc.

The typographical errors have been corrected.

  • Correct spelling of ammolactone-A throughout the text.

The manuscript has been checked for ammolactone-A (lines 352, 375)

Round 2

Reviewer 1 Report

The manuscript was revised according to the remarks and improved.